# Muscle Dysfunction and Functional Status in COVID-19 Patients during Illness and after Hospital Discharge

**DOI:** 10.3390/biomedicines12020460

**Published:** 2024-02-19

**Authors:** Otakar Psenicka, Tomas Brutvan, Jan Kratky, Jarmila Krizova

**Affiliations:** 13rd Department of Internal Medicine, General University Hospital, 120 00 Prague, Czech Republic; tomas.brutvan@vfn.cz (T.B.); jan.kratky@vfn.cz (J.K.); jarmila.krizova@vfn.cz (J.K.); 21st Faculty of Medicine, Charles University, 120 00 Prague, Czech Republic

**Keywords:** COVID-19, critical illness, long-term outcomes, muscle ultrasound, quality of life

## Abstract

Background: COVID-19 pneumonia is associated with SIRS and hypercatabolism. The aim of this study was to determine muscle loss during the acute phase of COVID-19 pneumonia and evaluate long-term sequelae in discharged patients. Methods: A total of 16 patients with COVID-19 pneumonia and respiratory insufficiency were included in the study. Selected parameters (weight, BMI, LBM = lean body mass, albumin, CRP, NLR = neutrophil-to-lymphocyte ratio, ultrasound measured thickness of rectus femoris muscle = US RF and rectus femoris + vastus intermedius = US RF + VI, handgrip strength, quality of life = EQ-5D questionnaire, and activities of daily living = Barthel’s ADLs) were recorded on admission, discharge, and 1, 3, and 6 months after discharge. Results: The most significant changes were between hospital admission and discharge: US RF and RF + VI (−1.28 ± 1.97 mm, *p* = 0.046; −1.76 ± 2.94 mm, *p* = 0.05), EQ-5D score (14.6 ± 19.2, *p* = 0.02), and ADLs (17.1 ± 22.6; *p* = 0.02). There was a significant positive correlation between US RF + VI and handgrip strength (*p* = 0.014) and a negative correlation between weight and Barthel index (*p* = 0.012). There was an association between muscle function with an EQ-5D score and ADLs during outpatient check-ups, most noticeably between handgrip strength, US RF+VI, and ADLs (*p* = 0.08; *p* = 0.1, respectively). **Conclusions:** In patients with COVID-19 pneumonia, there is a significant reduction of health-related quality of life, impaired even 6 months after hospital discharge, influenced mainly by muscle loss. During the hospital stay, there was a significant muscle mass reduction. Ultrasound measurement of thigh muscle thickness may be a useful method to monitor muscle loss.

## 1. Background

During the prior few years, a wide range of COVID-19-related symptoms and long-term consequences of the novel coronavirus infection have been observed [1]. While most patients can be asymptomatic or have only mild flu-like symptoms, severe forms, including COVID-19 pneumonia, are associated with systemic inflammation [2]. Activation of both innate and adaptive immune responses leads to elevated levels of proinflammatory cytokines, glucocorticoids, and catecholamines [3]. These factors among other complications cause hypercatabolism and breakdown of muscle proteins. Resulting muscle atrophy leads to increased mortality and longer lengths of hospital stays and may further cause long-term consequences referred to as post-COVID syndrome (also “long COVID” or “persistent COVID”) [4].

Frequent symptoms of post-COVID syndrome are fatigue, dyspnea, breathing discomfort, or cough but also include a broad spectrum of neuropsychiatric and cognitive disorders such as anxiety, depression, impaired concentration, memory, or sleep [5]. In general, these patients often have poor quality of life and reduced self-sufficiency [6].

Similar to other acute diseases, severe COVID-19 can lead to significant weight loss [7] and acute sarcopenia [8] with subsequent consequences such as overall functional decline, higher rate of falls and fractures, higher incidence of hospital admission, and also higher mortality rate [9]. Sarcopenia is associated with low muscle quantity and quality. Many methods of muscle mass measurement (e.g., bioelectric impedance BIA, dual-energy X-ray absorptiometry DXA, computed tomography CT, or magnetic resonance imaging MRI) are dominantly used in research studies, yet these parameters are not routinely investigated in clinical practice [10,11].

To assess the degree of muscle loss, an ultrasound measurement of thigh muscle thickness can be used. Skeletal muscle ultrasound is currently widely performed to assess muscle mass and/or muscle quality [12,13]. Due to its routine availability in most intensive care units, ultrasound investigation is an easy-to-perform non-invasive examination of sarcopenia [14] and intensive care unit-acquired weakness (ICUAW) [15]. In addition, the assessment can be performed repeatedly during the course of critical illness with excellent intra and interrater reliability when performed by a range of providers with no prior ultrasound experience [16]. The decline of muscle thickness correlates with ICUAW and with poor clinical outcomes [17]. To date, only a few studies of muscle ultrasound in COVID-19 patients have been published [18,19]. There was a significant correlation between reduction of rectus femoris thickness and higher mortality in ICU [18], and also with muscle strength one month after hospital discharge [19].

The aim of the present study was to determine muscle loss during the acute phase of COVID-19 pneumonia and evaluate long-term sequelae in discharged patients including their health-related quality of life. We also aimed to determine which of the monitored parameters are related to the patient’s functional status and quality of life in different stages of the disease. To the best of our knowledge, no studies comprehensively assessing the relationship between muscle wasting and quality of life in long-term periods in COVID-19 patients have yet been published. This study was conducted as a pilot study during the COVID-19 pandemic. Further studies of muscle loss-related changes in a wider spectrum of patients with critical illness are planned in our department.

## 2. Methods

### 2.1. Data Collection, Patients—Ethics, Enrolment Criteria

This was a single-center prospective observational study conducted in a university hospital in Prague, Czech Republic. The data were collected during a long period of COVID-19-related hospitalizations and subsequent outpatient check-ups from December 2020 to October 2021. All patients admitted to the standard ward or to the intensive-care unit with COVID-19 infection confirmed with PCR test were considered for enrolment.

The inclusion criteria were infiltrate present on chest X-ray or CT (computerized tomography) scan and respiratory insufficiency requiring oxygen therapy to maintain oxygen saturation (SpO_2_) > 92%. The exclusion criteria were length of hospital stay less than 5 days, any severe condition interfering with conduction of in-hospital and/or outpatient tests and measurements (e.g., immobility, dementia, generalized malignancy, etc.) Written informed consent was obtained from all enrolled patients. Ethical approval was provided by the General University Hospital in Prague Ethics Commission.

### 2.2. Study Protocol

A total of 16 patients with COVID-19 pneumonia and respiratory insufficiency were included in the study from December 2020 to March 2021. Subsequent outpatient check-ups were conducted till October 2021. One patient died during hospitalization, one patient refused to continue the follow-up and two patients were excluded due to incomplete outpatient examinations. A sample size calculation was not performed in this study due to its pilot-feasibility study design.

The diagnosis of the present COVID-19 infection was established by RT-PCR positivity of nasopharyngeal swab according to standard hospital procedures. The diagnosis of pneumonia was based on radiological findings consistent with interstitial pneumonia on a chest X-ray or CT (computerized tomography) scan. Respiratory insufficiency was defined as a need for oxygen therapy to maintain oxygen saturation (SpO_2_) above 92 percent.

All included patients underwent clinical evaluation consisting of medical history, physical examination, and routine laboratory tests. The first appearance of symptoms related to COVID-19 infection and date of positive PCR test, weight and BMI, CRP, albumin, prealbumin, cholinesterase, and NLR (neutrophile to leukocytes ratio) on the admission date was recorded for further evaluation. Within 48 h from the admission, patients underwent bioimpedance analysis of the body composition with Bodystat Quadscan 4000 to establish the lean body mass (LBM). In the first two days of hospitalization, hand grip strength of the dominant hand with electronic hand dynamometer (Trailite TL-LSC 100) was obtained. The highest value of three measurements was used for analysis. Specific questionnaires assessing the health-related quality of life (EQ-5D questionnaire) and the Barthel index for activities of daily living (ADLs) were filled out with the patients according to the validated user guide. Premorbid quality of life was also collected from every patient for further evaluation of its long-term impairment. Ultrasound measurement of the right rectus femoris muscle (see forthcoming description) was also recorded in all enrolled patients. These measurements and methods were carried out in the first two days from hospital admission, on the day of hospital discharge, and then during the follow up 1, 3, and 6 months after hospital discharge. These outpatient check-ups were held in the consulting room next to the hospital ward and all examinations were performed on the same devices.

### 2.3. Ultrasonography

Ultrasound measurements were conducted with ultrasound UGEO HM70A (Samsung, Seoul, Korea). Linear transducer (3–16 MHz) in musculoskeletal pre-set was used to perform the measurement. Quadriceps measurements using diagnostic ultrasound were performed as previously described by Martín et al. [20] was used for quadriceps muscle measurement. Patients were lying supine with extended knees and toes pointing to the ceiling. The quadriceps femoris muscle of the dominant lower limb was used for the investigation. As the exact point for the measurement, the one-third of the distance from cranial patella margin to anterior inferior iliac spine on the pelvis was determined. We used a permanent marker to ensure the measurement was made at the same point each time. Three measurements of rectus femoris (RF) and vastus intermedius (VI) thickness were obtained and the average value was calculated and recorded.

### 2.4. Statistical Analysis

Statistical analysis was performed using STATISTICA 12 software (Statsoft Inc., Tulsa, OK, USA). The values of individual parameters were stated as Mean ± SD. To assess the significance of differences in the same patients in two monitored periods, a paired *t*-test was used. Correlations between changes in monitored parameters were assessed using Pearson correlation coefficients compiled into correlation matrix. All tests were performed on 5% level of significance.

## 3. Results

Table 1 presents baseline epidemiologic and anthropometric data. The mean age was 64 ± 9.5 and 81.3% were men. The mean BMI was 29.9 ± 2.4 kg m^−2^, 42% of patients were overweight, and 58% were obese. The length of hospital stay was 15.9 ± 7.1 days and the delay from COVID-19 symptoms onset to hospital admission was 10.6 ± 3.4 days.

### 3.1. Change of Monitored Parameters during Hospitalization and Follow-Up

The majority of monitored parameters have changed significantly between the day of hospital admission and discharge. Hand grip strength and albumin levels have changed significantly between hospital discharge and the first outpatient check-up.

Table 2 and Figure 1 show the progression of the monitored parameters at different stages of the illness.

Compared to baseline, there was a significant reduction of weight (−3.3 ± 2.7 kg, *p* = 0.0016) and LBM (−2 ± 2.7 kg, *p* = 0.028) during hospitalization. The loss of muscle mass corresponded with these findings in both measured parameters (RF and RF + VI reduction −1.28 ± 1.97 mm, *p* = 0.046 and −1.76 ± 2.94 mm, *p* = 0.05, respectively). On the other hand, the hand grip strength was similar on day 1 and hospital discharge (35.1 ± 8.9 N and 36.5 ± 6.7 N), with the biggest change between hospital discharge and first outpatient check-up (+3.1 ± 4.4 N; *p* = 0.03). After treatment of the disease, there was also a significant increase in EQ-5D score (14.6 ± 19.2, *p* = 0.02) and Barthel index (17.1 ± 22.6; *p* = 0.02). There were very high baseline levels of CRP (128.7 ± 51.6 mg/L) with a significant decrease during treatment (−113 ± 61.2 mg/L), similar development in NLR (−4.7 ± 3.2) was observed.

### 3.2. Correlation between Monitored Parameters

In the course of the illness, there was a significant positive correlation between the change of LBM and hand-grip strength between day 1 and hospital discharge (*p* = 0.001), Figure 2. A significant negative correlation between weight loss and change in Barthel index between day 1 and hospital discharge (*p* = 0.016) was also observed, Figure 3. During outpatient check-ups, there was a significant positive correlation between US RF + VI and hand grip strength (*p* = 0.002, *p* = 0.012 and *p* = 0.014, respectively), Figure 4, and we also observed a trend toward correlation of both of these parameters with LBM (*p* = 0.13, and *p* = 0.09, respectively).

There was an association of muscle function with EQ-5D score and Barthel index during outpatient check-ups, most significantly between handgrip strength, US RF + VI, and Barthel index (*p* = 0.08; *p* = 0.1, respectively), which would be very likely statistically significant with a larger number of subjects.

## 4. Discussion

To the best of our knowledge, none of the studies published so far in COVID-19 patients described such a wide range of parameters and their development over a six-month horizon after hospital discharge. Even with a small sample size, the study gave us an in-depth insight into their muscle mass and function development and long-term impairment of health-related quality of life.

Severe impairment of health-related quality of life (QoL) in COVID-19 pneumonia patients was observed and even six months after hospital discharge the QoL was not completely restored to premorbid levels. This finding corresponds with previously published results of large observational studies. In Wuhan, the longitudinal cohort study there was lower QoL, worse exercise capacity, more mental health abnormality, and increased healthcare use after discharge, observed even after 2 years in patients with long COVID symptoms [4].

Significant muscle wasting occurs early in critically ill patients [21]. In the present study, we observed a significant weight reduction (more than 3 kg during hospitalization) with a corresponding decrease in LBM and diameter of quadriceps femoris.

While using a single method to monitor the loss of skeletal muscles in critical care may be less reliable, a comprehensive assessment of multiple parameters can be useful. Ultrasound muscle investigation and bioimpedance analysis can be affected by fluid overload [22,23]. Hand grip strength measured with a handheld dynamometer can be less reliable in critically ill patients in the low range of strength [24,25,26]. However, our study showed an overall good correlation of these parameters with each other. Therefore, an ultrasound examination of the quadriceps femoris muscle may be a useful method to extend the muscle status monitoring.

A correlation between ultrasound-measured muscle wasting and functional status was shown in other studies [27]. Although intensive-care unit-acquired weakness is nowadays a well-known phenomenon in critically ill patients, the COVID-19 pandemic has shown us, that even patients with milder forms of this viral infection can suffer from long-term muscle weakness and other related symptoms interfering with their daily activities [4,5,6,28]. In overall very complicated pathophysiology of both critical state and COVID-19-related impairments, the relationship between muscle mass, muscle function, and quality of life of affected individuals is not linear and the assessment of only one variable may be of limited clinical value. The assessment of multiple parameters of muscular function, body composition, and validated questionnaires represents a strength of this study.

As many clinical studies have already shown, mortality does not increase with increasing BMI in critically ill patients [29,30,31]. On the other hand, in the case of COVID-19 infection, obesity, and being overweight are independent risk factors for developing severe COVID-19, need of mechanical ventilation, and overall poor outcomes [4,32]. In our study, we observed a significant negative correlation between weight (and BMI) and ADLs score, and an association with decreased QoL was also observed. This supports previous findings that overweight and obese patients can suffer from worse COVID-19 pneumonia outcomes.

Surprisingly, in our study, laboratory parameters did not overall correlate with selected outcomes. As shown by previous studies and meta-analyses, high levels of CRP are often observed in COVID-19 infection [33] and its levels are associated with disease severity [34]. High CRP levels in COVID-19 patients, normally lacking in other viral infections, can be caused by Macrophage Activating Syndrome [35,36]. Mean CRP levels in our patients were 128.70 mg/L ± 51.6, which corresponds with previous studies of patients with severe COVID-19 infection [37]. However, in our patients, CRP did not correlate with QoL score or ADLs nor with other monitored outcomes. As we can see from the mean albumin level at hospital admission (28.52 ± 4) and from the delay from symptoms onset to hospital admission (10.6 ± 3.4 days), our patients came to the hospital after a long period of struggle at home. In these patients, other factors than baseline CRP levels were essential for their future functional outcome, as we discussed above. The only laboratory parameter with a weak negative association with EQ-5D and ADLs score was NLR (neutrophil-to-lymphocyte ratio).

The single most representative factor determining later sequelae of COVID-19 pneumonia patients was not established in our study. Although we did not observe a significant correlation between directly measured muscle parameters (muscle ultrasound, hand grip strength) and patients’ functional status (EQ-5D score, Barthel index) due to the small sample size, the association would be very likely statistically significant with a larger number of probands.

These partial associations of various clinical and laboratory features demonstrate the importance of comprehensive assessment of COVID-19 patients to target specific interventions (intensive nutrition support, more intensive physiotherapy) at-risk groups of patients. Further studies are needed to better identify these at-risk patients. This was a pilot hypothesis-generating study.

## 5. Limitations

Our study has several limitations. The major limitation is a small sample size. Because of the challenging conditions during the pandemic time and the relatively rapid decline in the occurrence of severe COVID-19 infection after vaccination, we were unable to collect a larger number of patients. Since the sample size is small, it would have been beneficial to add age- and sex-matched critically ill patients with other serious illnesses for comparison. This is planned as a part of the following clinical trials of our research team. The limitation in terms of a single-center study could be seen rather as an advantage in the consistency of measurements (muscle ultrasound, etc.).

Another problematic limitation of our study is the relative heterogeneity of included patients. Some of them only had a few minor comorbidities, on the other hand, there were patients with severe associated diseases and complications. Due to the small sample size, it was impossible to distinguish between these conditions in terms of long-term sequelae in discharged patients. Also, we were unable to evaluate the potential effect of therapeutic interventions, such as remdesivir or glucocorticoids administration. In addition, other critical illnesses can produce similar long-term consequences as muscle weakness and impaired quality of life. Adding a control arm of critically ill patients with other illnesses (e.g., community-acquired pneumonia) would be of advantage to deepen knowledge in this area and it is planned in further research.

In the case of performed measurements, we used the proposed algorithms which should ensure good reproducibility. However, in femoral muscle ultrasound, we did not unfortunately use cross-sectional area measurement of rectus femoris (RF_CSA_). In later studies, it turned out that especially RF_CSA_ correlated best with selected clinical outcomes [38,39].

## 6. Conclusions

Despite these limitations, femoral muscle ultrasound, handheld dynamometer, and bioimpedance analysis are non-invasive, easy-to-perform, and economically feasible methods with good reproducibility and great inter- and intra-observer reliability. Using these investigations gave us deep insight into the development of patients’ muscle mass and function during a long period of COVID-19 pneumonia and its further consequences.

We demonstrated a significant reduction in health-related quality of life, which occurred rapidly after the onset of the illness and was impaired even 6 months after hospital discharge. There was a significant association between selected muscle parameters (muscle ultrasound, hand grip strength, lean body mass). Although we were unable to show a significant correlation between muscle wasting and subsequent quality of life impairment due to the small sample size, the trend toward association was clear. This was a pilot study and results from this study should be seen as hypothesis-generating and could inspire further larger observational or interventional studies. Muscle wasting can be seen as an important risk factor for further deterioration of patients’ health-related quality of life.

## Figures and Tables

**Figure 1 biomedicines-12-00460-f001:**
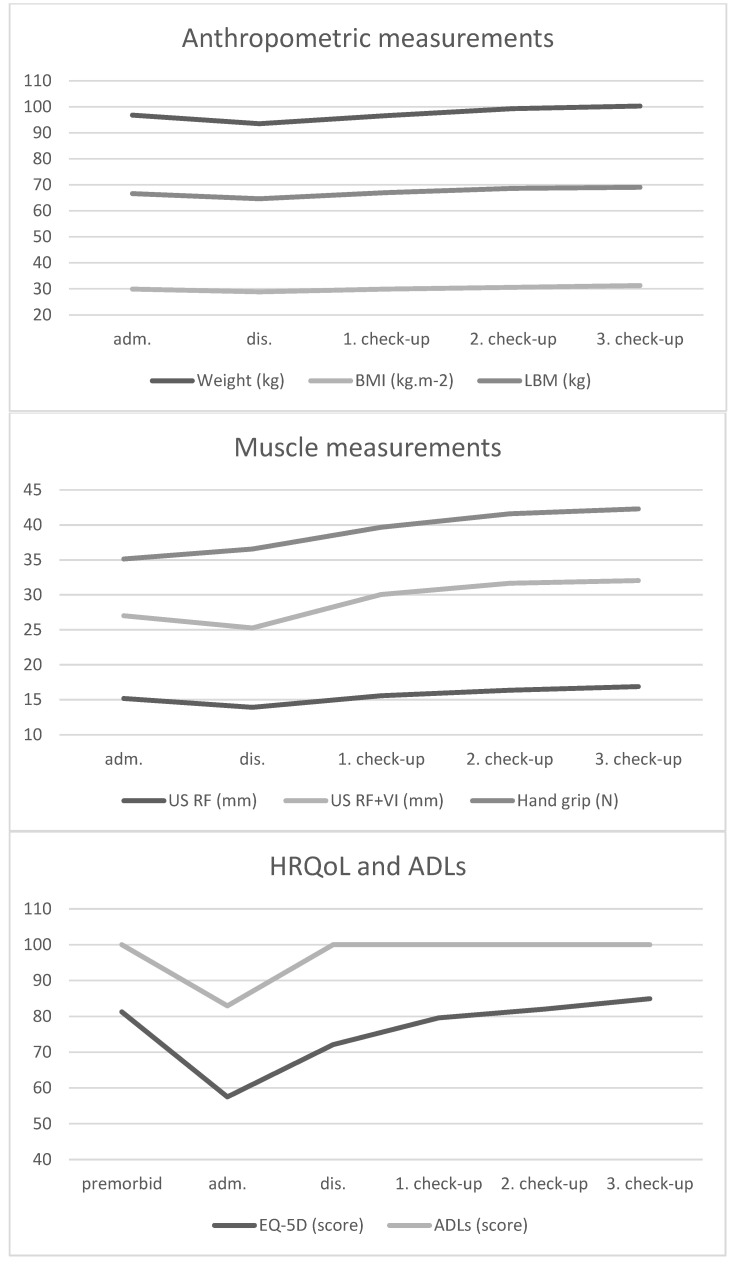
Values of monitored parameters during the course of illness. Abbreviations are discussed in Table 1 and Table 2.

**Figure 2 biomedicines-12-00460-f002:**
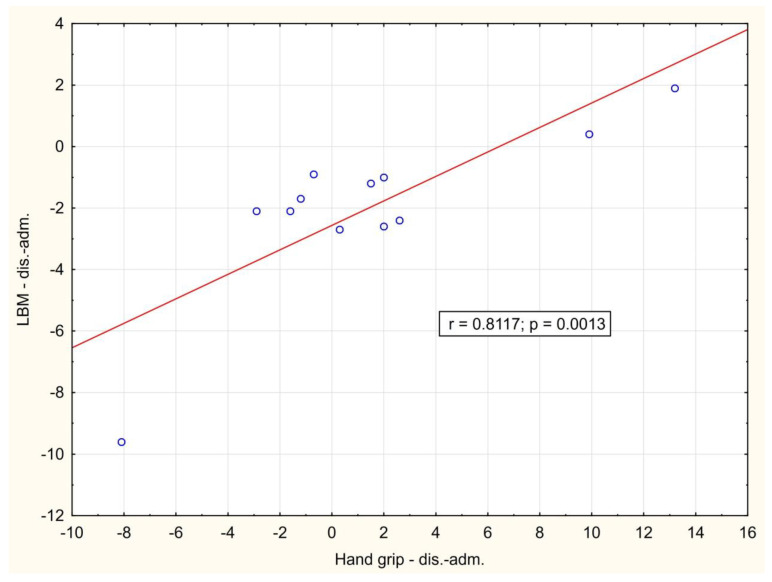
Correlation of the change in hand grip strength and lean body mass between hospital admission and hospital discharge. adm.: admission, dis.: discharge.

**Figure 3 biomedicines-12-00460-f003:**
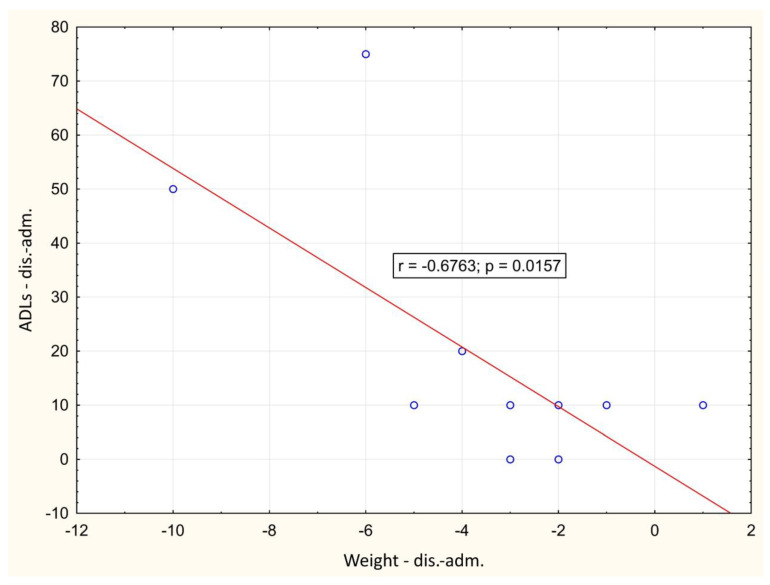
Correlation of the change in weight and Barthel index between day 1 and hospital discharge. adm.: admission, dis.: discharge.

**Figure 4 biomedicines-12-00460-f004:**
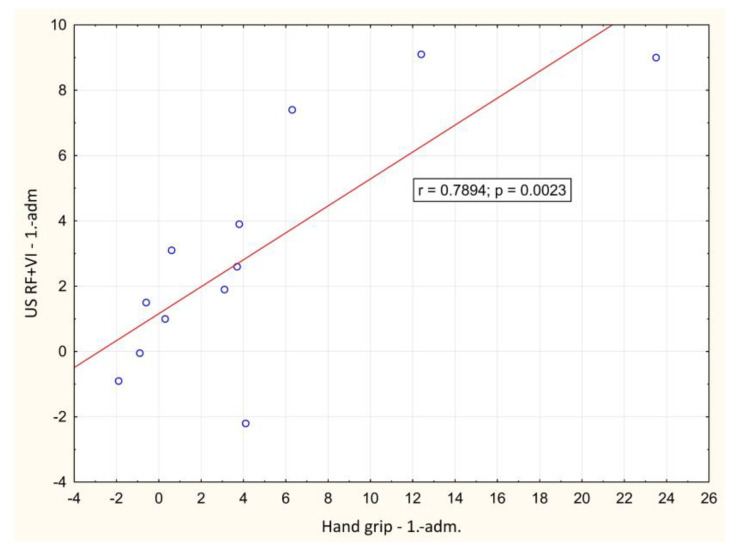
Correlation of the change in ultrasound measured rectus femoris + vastus intermedius thickness and hand grip strength value between baseline and outpatient check-ups. adm.: admission, dis.: discharge. 1.: 1st outpatient check-up, 2: 2nd outpatient check-up, 3rd outpatient check-up.

**Table 1 biomedicines-12-00460-t001:** Patients’ baseline characteristics.

	All Patients (*n* = 16)
Sex	Male (*n* = 13), Female (*n* = 3)
Age (years)	64 ± 9.5
Length of hospital stay (days)	15.9 ± 7.1
Delay from symptoms onset to hospital admission (days)	10.6 ± 3.4
Weight (kg)	96.83 ± 10.2
BMI (kg m^−2^)	29.93 ± 2.4
Hand grip strength (N)	35.13 ± 8.9
US RF (mm)	15.19 ± 3.9
US RF+VI (mm)	27.01 ± 6.3
LBM (kg)	66.66 ± 8.7
EQ-5D	57.50 ± 20.1
ADLs	82.92 ± 22.6
CRP (mg/L)	128.70 ± 51.6
Albumin (g/L)	28.52 ± 4
NLR	8.03 ± 2.8

US RF—ultrasound diameter of rectus femoris, US RF+VI—ultrasound diameter of rectus femoris + vastus intermedius, LBM—lean body mass, EQ-5D score—Euro Quality of life score, ADLs—activities of daily living (Barthel index), NLR—neutrophil to leucocyte ratio.

**Table 2 biomedicines-12-00460-t002:** Average values of monitored parameters during the course of illness.

	AVG (adm.)	AVG (dis.)	AVG (1. Check-Up)	AVG (2. Check-Up)	AVG (3. Check-Up)
Weight (kg)	96.83 ± 10.2	93.50 ± 9.92 *	96.59 ± 9.58 *	99.33 ± 10.55	100.28 ± 10.82
BMI (kg m^−2^)	29.93 ± 2.4	28.88 ± 2.4 *	29.96 ± 2.17 *	30.65 ± 2.25	31.29 ± 2.74
Hand grip (N)	35.13 ± 8.9	36.55 ± 6.7	39.67 ± 6.9 *	41.59 ± 7.2	42.30 ± 7.0
US RF (mm)	15.19 ± 3.9	13.91 ± 2.8 *	15.55 ± 2.6 *	16.36 ± 2.5	16.87 ± 2.3
US RF + VI (mm)	27.01 ± 6.3	25.26 ± 5.3 *	30.04 ± 5.7 *	31.66 ± 5.5	32.04 ± 5.6
LBM (kg)	66.66 ± 8.7	64.66 ± 8.7 *	67.05 ± 8.4 *	68.66 ± 8.9	69.06 ± 9.5
EQ-5D	57.50 ± 20.1	72.08 ± 12.4 *	79.58 ± 8.6	82.00 ± 11.0	84.92 ± 8.6
ADLs	82.92 ± 22.6	100.00 ± 0 *	100.00 ± 0	100.00 ± 0	100.00 ± 0
CRP (mg/L)	128.70 ± 51.6	15.59 ± 26.2 *	3.18 ± 2.0	10.86 ± 30.9	6.90 ± 17.3
Albumin (g/L)	28.52 ± 4	28.52 ± 7.1	44.23 ± 6.5 *	46.39 ± 2.9	45.40 ± 2.4
NLR	8.03 ± 2.8	3.31 ± 2.6 *	1.68 ± 1.0	1.87 ± 0.9	1.93 ± 0.8

AVG: average, adm.: admission, dis.: discharge. Other abbreviations are discussed in Table 1. * Significant change from the previous value.

## Data Availability

The raw data supporting the conclusions of this article are available on request from the corresponding author.

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
