# Peer review of "Muscle Dysfunction and Functional Status in COVID-19 Patients during Illness and after Hospital Discharge"

_biomedicines, 2024, doi:10.3390/biomedicines12020460_

Round 1
Reviewer 1 Report
Comments and Suggestions for Authors
very well done study but as the author said too many limitations.
critical illness of any kind can produce similar results. it would have been better if there was a control arm (critically ill patients with another illness like CAP).
since the sample size is small, age and sex matched critically ill patients with other serious illnesses can be recruited for comparison.
the severity of illness or the severity of COVID could have been measured with inflammatory parameters and correlated clinically.
persistent / long covid could have contributed as all patients were critically ill/ in ICU.
Comments on the Quality of English Language
looks fine, some grammatical errors need to be addressed.
Author Response
Thank you very much for taking the time to review this manuscript and for your thoughtful insights and constructive feedback. I have carefully addressed each point of your review. Please find the detailed responses below and see the corresponding revisions highlighted in track changes in the revised manuscript.
Comment 1: Critical illness of any kind can produce similar results. it would have been better if there was a control arm (critically ill patients with another illness like CAP).
Response 1: Thank you very much for the comment. It provides an interesting topic for further research. We have added the idea to the discussion (section “limitations” respectively, lines 282-286 in the revised manuscript). Thank you once again for pointing this out.
Comment 2: Since the sample size is small, age and sex matched critically ill patients with other serious illnesses can be recruited for comparison.
Response 2: We agree with this point. Adding age- and sex- matched critically ill patients with other serious illnesses is planned as a part of the following clinical trials of our research team. We have added the idea to the discussion (section “limitations” respectively, lines 272-274 in the revised manuscript).
Comment 3: the severity of illness or the severity of COVID could have been measured with inflammatory parameters and correlated clinically.
Response 3: Thank you, we appreciate this comment. We have discussed this issue in the discussion section of revised manuscript, please see lines 247-250. We have not divided patients into groups based on COVID-19 severity due to the small sample size. We have recorded average CRP values during the course of illness (see Table 2), but as mentioned above, they did not corelated with selected outcomes.
Comment 4: persistent / long covid could have contributed as all patients were critically ill/ in ICU.
Response 4: Agreed, this point should be considered, and we are planning further studies to enlighten this problem. It is discussed in discussion section (lines 231-234)
Reviewer 2 Report
Comments and Suggestions for Authors
this study is interesting, and rather well described. The main limit of this article is its very small sample size, and its purely descriptive design.
line 3: add study country and study design
line 12: how these 16 patients are selected
line 13: add years
line 66: you have to say why it's a pilot? 1st stage of a larger study?
line 88: there has been a sample calculation and this sample is purely pragmatic?
Author Response
Thank you very much for taking the time to review this manuscript and for your thoughtful insights and constructive feedback. I have carefully addressed each point of your review. Please find the detailed responses below and see the corresponding revisions highlighted in track changes in the revised manuscript.
Comment 1: line 3: add study country and study design
Response 1: Agreed, we added study country (lines 5, 7 of the revised manuscript). Study design is discussed in section “methods” (lines 74-77)
Comment 2: line 12: how these 16 patients are selected
Response 2: Thank you for the comment. We have added “16 patients with COVID-19 pneumonia and respiratory insufficiency” in the abstract text (see lines 12-13 in the revised manuscript). Patient’s inclusion and exclusion criteria are mentioned in section “Methods” (lines 79-85) including the exact COVID-19 infection and respiratory insufficiency diagnostic criteria (lines 95-99)
Comment 3: line 13: add years
Response 3: Thank you very much for pointing this out. We have made major changes in our abstract and the selected line is no longer in the text.
Comment 4: line 66: you have to say why it's a pilot? 1st stage of a larger study?
Response 4: We agree, thank you for mentioning this issue. We have added the answer to your comment into the revised manuscript – please see lines 71-73. It is also discussed in “Limitations” section (lines 272-274 and 282-286) and “Conclusions” (lines 302-304)
Comment 5: line 88: there has been a sample calculation and this sample is purely pragmatic?
Response 5: Thank you for the comment. Since this is a pilot study, there was no sample calculation. We added this to the revised manuscript (lines 93-94). All patients admitted to the hospital were considered for enrolment, inclusion and exclusion criteria are mentioned in lines 81-86.
Reviewer 3 Report
Comments and Suggestions for Authors
Standard to use periods (.) instead of commas (,) when presenting decimals. Should change throughout
Should have ( ) after 1st time spelling out quad names so readers know subsequent abbreviations
L21: does LBM and hand-grip correlation mean anything? Should you be looking at percentage change? It would make sense that someone with more LBM might have more hand-grip strength – but that doesn’t really address anything of interest
L33: Last years is not proper grammar. During the prior few years…
L33: Lead sentence needs revision for proper English grammar
L38: complications
L42: dyspnea
L88: Subsequent not consequent
L97 4000 to establish the lean body mass (LBM).
L99: don’t need also
L105: don’t say see below. (see forthcoming description)
L114: needs revision for grammar
L123: was performed using STATISTICA….
Stats seem appropriate, however, was data normally distributed? If not, nonparametric tests would be needed
L139: needs grammar revision
L143: development of monitored parameters is an odd choice of words
L165: avoid first person language; change throughout
Discussion
It is not standard to write out topics 1, 2, 3, 4… etc These should be lead in sentences in each paragraph and then you discuss your findings
Also, it is not standard to repeat your results and stats. Discuss, but don’t repeat
Comments on the Quality of English LanguageSee aforementioned section
Author Response
Thank you very much for taking the time to review this manuscript and for your thoughtful insights and constructive feedback. I have carefully addressed each point of your review. Please find the detailed responses below and see the corresponding revisions highlighted in track changes in the revised manuscript.
Comment 1: Standard to use periods (.) instead of commas (,) when presenting decimals. Should change throughout.
Response 1: Thank you very much for pointing this out. We changed it throughout the manuscript.
Comment 2: Should have ( ) after 1st time spelling out quad names so readers know subsequent abbreviations
Response 2: Agreed, we revised this – please see lines 14-15 in the revised manuscript.
Comment 3: L21: does LBM and hand-grip correlation mean anything? Should you be looking at percentage change? It would make sense that someone with more LBM might have more hand-grip strength – but that doesn’t really address anything of interest
Response 3: Thank you for the comment. We deleted this sentence in the manuscript, since it is not an important result of our research. This correlation could be useful in situations, when only one of these investigation methods is available. But as you say, this correlation is logical and it does not address anything of interest.
Comment 4: L33: Last years is not proper grammar. During the prior few years…
Response 4: Agreed, we revised this (line 31)
Comment 5: L33: Lead sentence needs revision for proper English grammar
Response 5: Thank you, we revised the sentence (line 31 - During the prior few years we could observe a wide range of COVID-19 related symptoms and long-term consequences of novel coronavirus infection)
Comment 6: L38: complications
Response 6: Thank you for pointing out this error, we corrected it (line 36)
Comment 7: L42: dyspnea
Response 7: Thank you for pointing out this error, we corrected it (line 41)
Comment 8: L88: Subsequent not consequent
Response 8: Thank you for pointing out this error, we corrected it (line 36)
Comment 9: L97 4000 to establish the lean body mass (LBM).
Response 9: Thank you, we changed it according to your comment (line 106)
Comment 10: L99: don’t need also
Response 10: Agreed, “also” was deleted (line 106)
Comment 11: L105: don’t say see below. (see forthcoming description)
Response 11: Thank you for the comment, we changed it (line 113)
Comment 12: L114: needs revision for grammar
Response 12: Thank you, we changed the sentence throughout (line 121-122).
Comment 12: L123: was performed using STATISTICA….
Response 12: Thank you, we changed the sentence according to your comment (line 131)
Comment 13: Stats seem appropriate, however, was data normally distributed? If not, nonparametric tests would be needed.
Response 13: Thank you for the comment. Yes, data were normally distributed.
Comment 14: L139: needs grammar revision
Response 14: Thank you, we changed the sentence throughout (line 147-148)
Comment 15: L143: development of monitored parameters is an odd choice of words
Response 15: Agreed, we changed the sentence throughout (line 150)
Comment 16: L165: avoid first person language; change throughout
Response 16: Thank you, we changed the sentence (line 173)
Comment 17: It is not standard to write out topics 1, 2, 3, 4… etc These should be lead in sentences in each paragraph and then you discuss your findings
Response 17: Agreed, we changed the discussion section throughout according to your comment. (please see the discussion in the revised manuscript)
Comment 18: Also, it is not standard to repeat your results and stats. Discuss, but don’t repeat
Response 18: Thank you for the comment. We deleted most of the repeated results in the discussion. The only remaining values in the discussion section are mean CRP and albumin levels, because we think these results are necessary to discuss laboratory parameters and their eventual correlation with selected outcomes.
Round 2
Reviewer 2 Report
Comments and Suggestions for Authors
thanks to the authors for additions and corrections, this article seems to me to be publishable in this form
Reviewer 3 Report
Comments and Suggestions for Authors
I like that the authors discussed the pilot nature of this study. Consider adding to title
L31: avoid 1st person language like "we"
L72: was this part of a larger study? If not, I don't think you need to state "prior to.... critical illness". I think you could just leave it as pilot study
L121: The reference for quad measurement seems odd. Maybe it is a good article, I would rephrase like this: Quadriceps measurements using diagnostic US was performed a previously described by Martin et al (20).
L148: Merely seems like an odd word here. Not sure what is being said
Table 2: recommend spelling out abbreviations below the table
You did a good job changing , to .; however this needs to be doing in your figures also
L214: avoid conversational language like "Nowadays it is widely known.."
Comments on the Quality of English LanguageSee above